# CXPMRG-Bench: Pre-training and Benchmarking for X-ray Medical Report Generation on CheXpert Plus Dataset

## Abstract

X-ray image-based medical report generation (MRG) is a pivotal area in artificial intelligence which can significantly reduce diagnostic burden and patient wait time. Despite significant progress, we believe that the task has reached a bottleneck due to the limited benchmark datasets and the existing large models' insufficient capability enhancements in this specialized domain. Specifically, the recently released CheXpert Plus dataset provides only the dataset itself, lacking comparative evaluation algorithms and their performances. This situation makes the training, evaluation, and comparison of subsequent algorithms challenging. Thus, we conduct a comprehensive benchmarking of existing mainstream X-ray report generation models and Large Language Models (LLMs), on the CheXpert Plus dataset. We believe that the proposed benchmark can provide a solid comparative basis for subsequent algorithms and serve as a guide for researchers to quickly grasp the state-of-the-art models in this field. More importantly, we propose a new large model for X-ray image report generation by using a multi-stage pre-training strategy which involves Mamba based self-supervised autoregressive generation, X-ray-report contrastive learning, and supervised fine-tuning. Extensive experimental results demonstrate that the proposed Mamba based autoregressive pre-training can effectively encode X-ray images, and the image-text contrastive pre-training further aligns the feature spaces, achieving better experimental results. All the source codes will be released upon acceptance.

## 1 Introduction

X-ray image based Medical Report Generation (MRG) is a critical research problem in artificial intelligence, which targets describing the *findings* or *impressions* from the given X-ray data via natural language. The successful implementation of this task can significantly reduce the diagnostic burden on physicians, decrease patient wait time, and foster the positive application of artificial intelligence. However, the path to progress in this direction is not smooth sailing, there remain formidable challenges that need to be addressed. The challenging issues include image interpretation, data annotation, heterogeneity issues, consistency and standardization of reports, diversity and variability of diseases, interpretability of algorithms, etc. How to address these challenges and further improve the quality of medical report generation remains an urgent research problem.

After revisiting the mainstream algorithms of X-ray image medical report generation, we find that datasets like IU X-ray and MIMIC-CXR are widely used for the training and evaluation of report generation models. However, the IU X-ray only contains 7,470 images and 3,955 radiology reports samples, which is rather limited, especially in the large model era. The recently released CheXpert Plus dataset Chambon et al. (2024) is a large-scale dataset for the X-ray report generation, however, they did not release comparative methods, making it difficult for subsequent algorithms to conduct experiments and comparisons on this dataset. Therefore, we conduct a comprehensive benchmarking of existing open-sourced mainstream X-ray report generation models, Large Language Models (LLMs), and Vision-Language Models (VLMs), termed **CXPMRG-Bench**, on the newly released CheXpert Plus dataset, as shown in Fig. 1. The completion of this work can also help researchers identify which large models and algorithms are currently leading in the field of X-ray report generation.

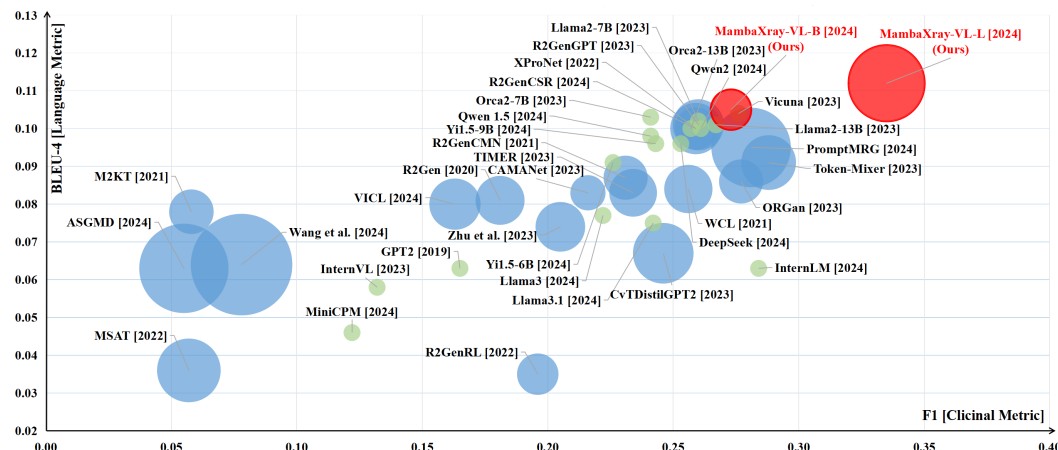

Figure 1: An overview of the benchmarked LLM/VLM-based (green circle) and mainstream MRG models (blue circle) on the CheXpert Plus dataset in this paper. For mainstream MRG models and our algorithms, the area of the circle represents the number of parameters.

On the other hand, most mainstream algorithms follow the encoder-decoder framework which usually adopts the vision encoder (e.g., ResNet He et al. (2016), Transformer Vaswani et al. (2017)) to process the given X-ray data and a text decoder (e.g., LSTM Hochreiter & Schmidhuber (1997), GRU Chung et al. (2014), Transformer Vaswani et al. (2017)) for report generation. Along with the development of pre-trained LLM and VLM, the quality of medical reports is enhanced significantly. There are already some researchers who exploit the pre-training for the X-ray report generation. For example, Wang et al. Wang et al. (2024c) propose high-definition X-ray vision models using context-aware masked auto-encoder. CXR-CLIP You et al. (2023) is a new pre-training method that generates more image-text pairs and introduces contrastive loss to enhance the discriminative power of images and texts, effectively learning features in the CXR domain. PTUnifier Chen et al. (2023c) proposes a simple and effective method that utilizes visual and textual prompt pools to make the model compatible with different types of inputs, thereby unifying the advantages of fusion encoders and dual encoders. However, we believe these models may be limited by the following issues: *Firstly*, the Transformer vision backbone brings huge computational costs $\mathcal{O}(N^2)$, which is not hardware friendly; *Secondly*, pure X-ray images are abundant and readily collectible, but paired X-ray and report data are relatively scarce. Current X-ray models are pre-trained in a single stage using X-ray image or image-report data only, and fail to exploit the full potential of these diverse X-ray data, therefore, the overall performance may be limited.

To address the issues mentioned above, in this work, we exploit multi-stage pre-training for the X-ray image MRG task and propose the **MambaXray-VL** large model, including *self-supervised autoregressive generation* and *Xray-report contrastive learning*, and *supervised fine-tuning* on the downstream report generation datasets, as shown in Fig. 2. Specifically speaking, we first partition and feed the X-ray image into the Mamba network to predict the next tokens based on previous context tokens in an autoregressive generation manner. This will enhance the vision perception ability of X-ray significantly using the relative low-cost Mamba network ($\mathcal{O}(N)$). For the second stage, we feed the paired X-ray image and corresponding report into the Mamba vision backbone and text encoder (Bio_ClinicalBERT Alsentzer et al. (2019), Llama2 Touvron et al. (2023)) for contrastive learning. It will align the X-ray image and reports using the pre-trained feature space. After that, we conduct supervised fine-tuning on each downstream X-ray report generation dataset to achieve higher performance by feeding the X-ray image into the pre-trained Mamba vision backbone network and LLM decoder network. Extensive experiments on three MRG benchmark datasets demonstrate that our pre-trained MambaXray-VL model achieves state-of-the-art performance.

To sum up, the contributions of this paper can be summarized as the following three aspects:

1). We conduct a comprehensive benchmark for the newly released CheXpert Plus dataset Chambon et al. (2024), termed **CXPMRG-Bench**, which covers 19 mainstream X-ray medical report

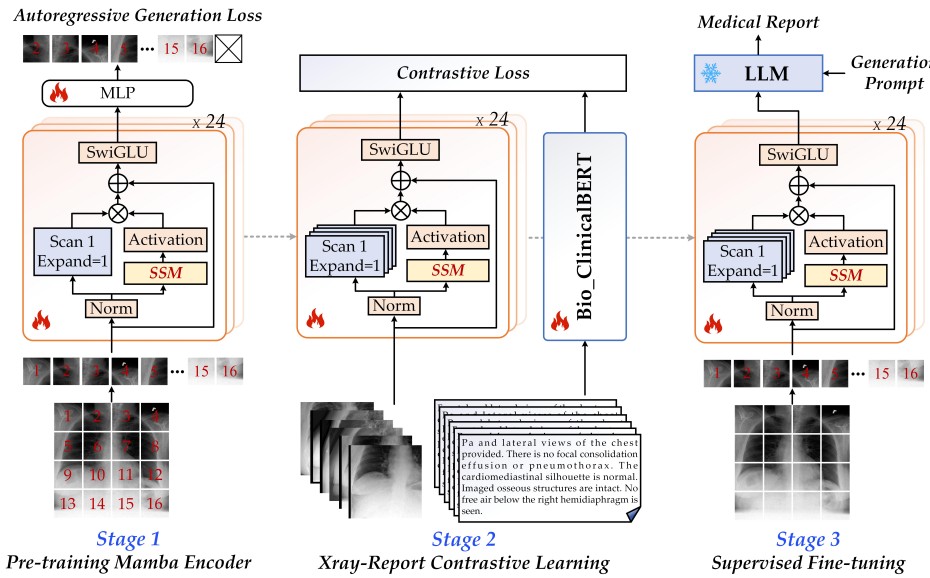

Figure 2: An overview of our proposed MambaXray-VL pre-training framework. It contains three training stages, i.e., Mamba-based autoregressive generation, Xray-report based contrastive learning, and supervised fine-tuning. Note that the layers or modules with *fire/snow* symbols denote the parameters that are tuned/frozen in the training phase.

generation algorithms, 14 large language models, and 2 vision-language models. To the best of our knowledge, this benchmark is the first large-scale evaluation of the CheXpert Plus dataset, providing subsequent researchers in the field of X-ray report generation with important reference and comparison criteria.

2). We propose a new pre-trained large model, termed **MambaXray-VL**, which adopts the Mamba as the vision encoder and the large language model as the text decoder. Compared to the Transformer, the proposed Mamba-based framework achieves state-of-the-art performance with lower computational cost via the multi-stage pre-training strategy.

3). We extend our research to a broader scope by conducting experiments on the IU X-ray and MIMIC-CXR datasets. We perform analytical experiments and visualizations to deepen the understanding of our MambaXray-VL model's performance and its capabilities in generating X-ray medical reports.

## 2 MAMBAXRAY-VL LARGE MODEL

### 2.1 OVERVIEW

As shown in Fig. 2, we propose a new multi-stage pre-training strategy for the X-ray image medical report generation, including *self-supervised autoregressive generation*, *Xray-report contrastive learning*, and *supervised fine-tuning*. The key insight of our multi-stage pre-training instead of joint training is that the aligned Xray-report data are limited, but there are more publicly available X-ray images. Thus, we first pre-train a large-scale vision backbone network on the X-ray images by using the Mamba layers, due to a better balance between the computational cost and accuracy. More importantly, we adopt the autoregressive generation to achieve self-supervised learning on the X-ray images. It performs similar or even better than the widely used MAE (Masked Auto-Encoder) pre-training strategy on this task. Then, we transfer the Mamba vision backbone to the second stage, i.e., Xray-report contrastive learning. Specifically, we feed the paired data into the pre-trained Mamba vision backbone and language encoder for the vision-language feature extraction. This stage will project the vision and language representations into a shared feature space to bridge the vision-

semantic gaps. Finally, we conduct supervised fine-tuning on the training subset of downstream datasets for the X-ray medical report generation.

## 2.2 MULTI-STAGE PRE-TRAINING

As illustrated in Fig. 2, our proposed *MambaXray-VL* large model contains three training stages which will be introduced in the following paragraphs respectively.

• **Stage #1: Auto-regressive Generation for Mamba Vision Encoder Pre-training.** To make full use of existing X-ray images, we conduct self-supervised learning to obtain a strong vision backbone network. Different from the widely used MAE (Masked Auto-Encoder)-based framework, in this work, we find that the autoregressive generation based framework works similarly or even better for the X-ray images, as inspired by the success of autoregressive generation in ChatGPT OpenAI (2023), GPT-4 Achiam et al. (2023), and ARM Ren et al. (2024). Let's denote the X-ray image as $\mathcal{I} \in \mathbb{R}^{192 \times 192 \times 3}$. We first partition it into non-overlapping image patches $\mathcal{P}_i \in \mathbb{R}^{16 \times 16 \times 3}, i = \{1, 2, ..., N\}$ and project them into visual tokens $\mathcal{T}_i \in \mathbb{R}^{1024}, i = \{1, 2, ..., N\}$ by using a convolutional layer (kernel size $16 \times 16$). Here, *N* is *144* when the resolution of the input X-ray image is set as $3 \times 192 \times 192$. Then, we feed the visual tokens into the *Vim* Zhu et al. (2024) backbone network for feature extraction whose complicity $\mathcal{O}(N)$ is much lower than the widely used Transformer $\mathcal{O}(N^2)$. The key operation of *Vim* is the Mamba block (a specific variation of State Space Model Wang et al. (2024d)), as shown in Fig. 2. The visual tokens will first be normalized and fed into the SSM and scan branches. The outputs will be multiplied and added with residual connections. The SwiGLU Shazeer (2020) is adopted to further process output features before being fed into subsequent Mamba blocks. Finally, a MLP layer is adopted for token reconstruction by using the auto-regressive generation loss function.

The objective of autoregressive pre-training is to predict the probability $p$ of the next token one by one based on the given corpus $\mathcal{T} = \{\mathcal{T}_1, \mathcal{T}_2, ..., \mathcal{T}_n\}$, which can be formulated as:

$$p(\mathcal{T}) = \prod_{i=1}^{n} p(\mathcal{T}_i | \mathcal{T}_1, ..., \mathcal{T}_{i-1}, \Theta). \tag{1}$$

We can find that the likelihood of each token $\mathcal{T}_i$ is computed based on the context of all the proceeding tokens $\{\mathcal{T}_1, ..., \mathcal{T}_{i-1}\}$. Thus, the loss function used for Stage #1 can be formulated as follows:

$$\mathcal{L}_{AR} = \sum_{i=1}^{n-1} |Vim([\mathcal{T}_1, ..., \mathcal{T}_i]) - \mathcal{T}_{i+1}|^2, \tag{2}$$

where $Vim$ denotes the vision Mamba backbone network.

• **Stage #2: Xray-Report Contrastive Learning.** We adopt the Mamba vision backbone network from the first stage and conduct contrastive learning based on mini-batch Xray-report samples. This will further align the dual modalities via the CLIP loss as validated in the CLIP Radford et al. (2021). In our implementation, we randomly sample a mini-batch and feed the X-ray images and medical reports into the Vim backbone and the language model (Bio_ClinicalBERT Alsentzer et al. (2019), Llama2 Touvron et al. (2023)) and compute the cosine similarity between the X-ray image and medical reports:

$$\mathcal{L}_{CTL} = CLIPLoss(Vim(\mathcal{I}_i), LM(\mathcal{R}_j)), \tag{3}$$

where $i$ and $j$ are the index of the X-ray image $\mathcal{I}$ and report annotation $\mathcal{R}$.

• **Stage #3: Supervised Fine-tuning.** After the above two pre-training stages, we conduct supervised fine-tuning on the training subset of X-ray image medical report generation. Similar to the first stage, we partition the given X-ray image into non-overlapping patches and project them into visual tokens. Then, the pre-trained Vim backbone network is used for the feature extraction. We concatenate the visual tokens and generation prompt as the input of a large language model for high-performance medical report generation.

In this stage, we adopt the *negative log-likelihood* as the loss function, i.e.,

$$\mathcal{L}_{NLL} = -\sum_{i=1}^{T} log p_\theta(y_i | Prompt, [y_1, ..., y_{i-1}]), \tag{4}$$

Table 1: Experimental Results on the CheXpert Plus Dataset using **Mainstream Medical Report Generation Algorithms**. **B4**, **R**, **M**, and **C** is short for BlEU-4, ROUGE-L, METEOR, CIDEr, respectively. **P**, **R**, and **F1** is short for Precision, Recall, F1 score, respectively. *min* is short for minutes. The parameter listed in this table denotes the parameters that need to be tuned in the training phase. The best result is highlighted in bold, and the second-best result is underlined.

| Index | Algorithm | Publish | Encoder | Decoder | B4, R, M, C | P, R, F1 | Time | Param | Code |
|---|---|---|---|---|---|---|---|---|---|
| #01 | R2GenRL | ACL22 | Transformer | Transformer | 0.035, 0.186, 0.101, 0.012 | 0.193, 0.229, 0.196 | 44.33 | 59.87 | URL |
| #02 | XProNet | ECCV22 | Transformer | Transformer | 0.100, 0.265, 0.146, 0.121 | 0.314, 0.247, 0.259 | 6.3 | 62.35 | URL |
| #03 | MSAT | MICCAI22 | ViT-B/16 | Transformer | 0.036, 0.156, 0.066, 0.018 | 0.044, 0.142, 0.057 | 5.72 | 141.10 | URL |
| #04 | ORGan | ACL23 | CNN | Transformer | 0.086, 0.261, 0.135, 0.107 | 0.288, 0.287, 0.277 | 46.66 | 67.50 | URL |
| #05 | M2KT | MIA21 | CNN | Transformer | 0.078, 0.247, 0.101, 0.077 | 0.044, 0.142, 0.058 | 22.5 | 69.07 | URL |
| #06 | TIMER | CHIL23 | Transformer | Transformer | 0.083, 0.254, 0.121, 0.104 | 0.345, 0.238, 0.234 | 26.5 | 79.28 | URL |
| #07 | CvT2DistilGPT2 | AIM23 | Transformer | GPT2 | 0.067, 0.238, 0.118, 0.101 | 0.285, 0.252, 0.246 | 13.93 | 128 | URL |
| #08 | R2Gen | EMNLP20 | Transformer | Transformer | 0.081, 0.246, 0.113, 0.077 | 0.318, 0.200, 0.181 | 110.05 | 83.5 | URL |
| #09 | R2GenCMN | ACL21 | Transformer | Transformer | 0.087, 0.256, 0.127, 0.102 | 0.329, 0.241, 0.231 | 66.08 | 67.70 | URL |
| #10 | Zhu et al. | MICCAI23 | Transformer | Transformer | 0.074, 0.235, 0.128, 0.078 | 0.217, 0.308, 0.205 | 10.03 | 85.95 | URL |
| #11 | CAMANet | IEEE JBH23 | Swin-Former | Transformer | 0.083, 0.249, 0.118, 0.090 | 0.328, 0.224, 0.216 | 23.08 | 43.22 | URL |
| #12 | ASGMD | ESWA24 | ResNet-101 Transformer | Transformer | 0.063, 0.220, 0.094, 0.044 | 0.146, 0.108, 0.055 | 87.37 | 277.41 | URL |
| #13 | Token-Mixer | IEEE TMI23 | ResNet-50 | Transformer | 0.091, 0.261, 0.135, 0.098 | 0.309, 0.270, 0.288 | 17.54 | 104.34 | URL |
| #14 | PromptMRG | AAAI24 | ResNet-101 | Bert | 0.095, 0.222, 0.121, 0.044 | 0.258, 0.265, 0.281 | 108.45 | 219.92 | URL |
| #15 | R2GenGPT | Meta-Rad.23 | Swin-Transformer | Llama2 | 0.101, 0.266, 0.145, 0.123 | 0.315, 0.244, 0.260 | 77.8 | 90.9 | URL |
| #16 | WCL | EMNLP21 | Transformer | Transformer | 0.084, 0.253, 0.126, 0.103 | 0.335, 0.259, 0.256 | 24.08 | 81.29 | URL |
| #17 | R2GenCSR | arXiv24 | VMamba | Llama2 | 0.100, 0.265, 0.146, 0.121 | 0.315, 0.247, 0.259 | 31.2 | 91.7 | URL |
| #18 | VLCI | arXiv24 | Transformer | Transformer | 0.080, 0.247, 0.114, 0.072 | 0.341, 0.175, 0.163 | 123.71 | 91.46 | URL |
| #19 | Wang et al. | arXiv24 | ViT | Llama2 | 0.064, 0.220, 0.110, 0.059 | 0.175, 0.099, 0.078 | 10.82 | 358.80 | URL |
| #20 | MambaXray-VL-B | Ours | MambaXray-VL | Llama2 | 0.105, 0.267, 0.149, 0.117 | 0.333, 0.264, 0.273 | 50.66 | 57.31 | URL |
| #21 | MambaXray-VL-L | Ours | MambaXray-VL | Llama2 | **0.112**, **0.276**, **0.157**, **0.139** | **0.377**, **0.319**, **0.335** | 55.18 | 202.32 | URL |

where $\theta$ denotes the trainable parameters and $T$ is the number of words that are predicted by the large language model. *Prompt* is the instruction prompt which is "*Generate a comprehensive and detailed diagnosis report for this chest X-ray image.*" used in our experiments.

## 3 CXPMRG-BENCH

In this paper, we benchmark the newly released CheXpert Plus dataset for the X-ray image based medical report generation. The mainstream MRG algorithms and large language models are listed in the following subsections. For the experimental results, please refer to Table 1, Table 2, and Fig. 1.

### 3.1 MAINSTREAM MRG ALGORITHMS

For the mainstream X-ray image MRG algorithms, as shown in Table 1, we train and test 21 open-sourced algorithms from year 2020 to year 2024. These models adopt the **CNN** (ORGan Hou et al. (2023), M2KT Yang et al. (2023), ASGMD Xue et al. (2024), Token-Mixer Yang et al. (2024), PromptMRG Jin et al. (2024)), **Transformer** (R2GenRL Qin & Song (2022), XProNet Wang et al. (2022a), MSAT Wang et al. (2022b), TIMER Wu et al. (2023), CvT2DistilGPT2 Nicolson et al. (2023), R2Gen Chen et al. (2020), R2GenCMN Chen et al. (2021), Zhu et al. Zhu et al. (2023), CAMANet Wang et al. (2024a), R2GenGPT Wang et al. (2023), WCL Yan et al. (2021), VLCI Chen et al. (2023a), Wang et al. Wang et al. (2024c)), and **Mamba** (R2GenCSR Wang et al. (2024b), MambaXray-VL-B, MambaXray-VL-L) as their vision backbone network, and utilize the LSTM, Transformer based model as the decoder network. Note that, the MambaXray-VL-B and MambaXray-VL-L are two models proposed in this paper which will be introduced in the next section. When reproducing these X-ray based MRG models, we found that some algorithms use *truncated ground truth* for comparison, which we believe may not accurately reflect the true evaluation results. Therefore, we abandoned the truncation mechanism and used the complete ground truth for result evaluation, making the obtained results more accurate and reliable.

### 3.2 LLMS FOR MRG

We evaluate a total of 16 open-source LLMs, as shown in Table 2, including Vicuna-7B Zheng et al. (2023), QWen1.5-7B et al. (2023), QWen2-7B-Instruct et al. (2023), InternLM-7B Cai et al. (2024), Llama2-7B Touvron et al. (2023), Llama2-13B Touvron et al. (2023), Llama3-8B Dubey et al. (2024), Llama3.1-8B Dubey et al. (2024), GPT2-Medium Radford et al. (2019), Orca 2-7B Mitra et al. (2023), Orca 2-13B Mitra et al. (2023), DeepSeek-LLM-7B-Chat Bi et al. (2024), Yi-1.5-6B-Chat Young et al. (2024), Yi-1.5-9B-Chat Young et al. (2024). Note that part of the LLMs is selected

from ***open-llm-leaderboard*** [1] and integrated with R2GenGPT Wang et al. (2023) model by replacing the Llama2 language decoder with corresponding LLMs. In our implementation, we keep the visual encoder SwinTransformer unchanged for a fair comparison. In addition, we also test two pre-trained vision-language large models, i.e., InternVL-2 Chen et al. (2023b) and MiniCPM-V2.5 Yao et al. (2024), to check whether a better performance can be obtained, as shown in Table 2.

### 3.3 EVALUATION RESULTS

**[Mainstream MRG Models]** As shown in Table 1, there are five MRG models which achieve a higher B4 metric, i.e., the XProNet Wang et al. (2022a) (0.100), R2GenGPT Wang et al. (2023) (0.101), R2GenCSR Wang et al. (2024b) (0.100), and our newly proposed MambaXray-VL-B and MambaXray-VL-L which achieves 0.105, and 0.112, respectively. It is intuitive to find that the large language model Llama2 works well for the MRG task. For F1 in the clinical metric, the top-5 models are our newly proposed MambaXray-VL-L (0.335), Token-Mixer Yang et al. (2024) (0.288), PromptMRG Jin et al. (2024) (0.281), ORGan Hou et al. (2023) (0.277) and our proposed MambaXray-VL-B (0.273). From these results, we can find that our proposed multi-stage pre-training strategy is rather effective in the disease-aware perception of the MRG.

**[LLM/VLM based MRG Models]** As shown in Table 2, we also report the performance of existing widely used LLMs by replacing the Llama2 based on the R2Gen-GPT framework (SwinTransformer is adopted as the vision backbone network). It is easy to find that the Vicuna-V1.5 Zheng et al. (2023) released in the year 2023 achieves the best B4 metric and the InternLM Cai et al. (2024) performs the best on the F1 clinical metric. For the two vision-language models we evaluated, i.e., the InternVL-2 and MiniCPM-V2.5, we can find that their results are not as good as other LLM-based models, although they have similar parameters. These results demonstrate that the vision-language models pre-trained on natural image-pairs may have large gaps with the X-ray medical images. Compared with the mainstream MRG models reported in Table 1, the LLM-based MRG achieves better results than regular language decoders which demonstrates the effectiveness of pre-trained LLMs.

**[Efficiency & Parameters]** From the perspective of running efficiency, we test these models on a server with A800 GPUs (80GB). Note that, we set the batch size as large as possible to make full use of the GPU memory. As a result, we can find that MSAT Wang et al. (2022b) and XProNet Wang et al. (2022a) are the first two algorithms that only need 5.72 and 6.3 minutes for the testing subset. R2Gen Chen et al. (2020), PromptMRG Jin et al. (2024), and VLCI Chen et al. (2023a) are relatively slow and need more than 100 minutes on the testing subset of CheXpert Plus dataset. For the LLM-based MRG reported in Table 2, we can find that Yi-1.5 Young et al. (2024) with 6.1B and 8.8B achieves better efficiency which needs 43.66 and 48.50 minutes for the testing. From the Fig. 1 and Table 1, we can find that the ASGMD Xue et al. (2024), PromptMRG Jin et al. (2024), Wang et al. Wang et al. (2024c), and our MambaXray-VL-L contains the most parameters (larger than 200M) needed to be tuned in the training phase. However, we can find that our model runs faster than these large models which only needs 55.18 minutes. It fully validates the efficiency of our proposed framework for the X-ray image based medical report generation.

## 4 EXPERIMENTS

### 4.1 DATASET AND EVALUATION METRIC

In the first stage of autoregressive pre-training, we used about 1.27 million medical chest X-ray images proposed in the work Wang et al. (2024c). In the second stage of image-text contrastive learning pre-training, we used a combination of training data from the **MIMIC-CXR** Johnson et al. (2019), **CheXpert Plus** Chambon et al. (2024), and **IU X-ray** Demner-Fushman et al. (2016) datasets, totaling 480k image-report pairs. Note that the CheXpert Plus dataset used here consists of images and impressions, not the image and findings combination used in the third stage. We strictly excluded any testing samples used in the third stage, resulting in a total of 210k image-impression pairs. In the third stage, we evaluate the performance of our model on three datasets, including IU X-Ray Demner-Fushman et al. (2016), MIMIC-CXR Johnson et al. (2019), and CheXpert Plus Chambon et al. (2024) dataset.

---

[1] https://huggingface.co/spaces/open-llm-leaderboard/open_llm_leaderboard

Table 2: Experimental Results of Medical Report Generation on the CheXpert Plus Dataset using different **LLMs and VLMs based on R2Gen-GPT**. The symbol † indicates that the model is a VLM. The Param listed in this table denotes the parameters of LLM/VLM.

| Index | LLM/VLM | Year | B4 | R | M | C | P | R | F1 | Time (*min*) | Param | Code |
|-------|---------|------|-----|-----|-----|-----|-----|-----|-----|------|-------|------|
| #01 | Vicuna-V1.5 | 2023 | **0.104** | **0.272** | 0.160 | 0.202 | 0.334 | 0.258 | 0.276 | 72.00 | 6.7B | URL |
| #02 | Qwen-1.5 | 2024 | 0.098 | 0.262 | 0.139 | 0.139 | 0.303 | 0.233 | 0.241 | 154.25 | 7.7B | URL |
| #03 | Qwen-2 | 2024 | 0.100 | 0.270 | 0.142 | 0.159 | 0.313 | 0.269 | 0.261 | 103.33 | 7.6B | URL |
| #04 | InternLM | 2024 | 0.063 | 0.207 | 0.136 | 0.104 | 0.307 | **0.274** | **0.284** | 294.00 | 7.3B | URL |
| #05 | Llama-2 | 2023 | 0.102 | 0.267 | 0.157 | 0.179 | 0.315 | 0.244 | 0.260 | 77.78 | 6.7B | URL |
| #06 | Llama-2 | 2023 | 0.101 | 0.269 | 0.160 | **0.214** | 0.321 | 0.254 | 0.267 | 116.42 | 13.0B | URL |
| #07 | Llama-3 | 2024 | 0.077 | 0.220 | 0.121 | 0.134 | 0.306 | 0.232 | 0.222 | 130.00 | 8.0B | URL |
| #08 | Llama-3.1 | 2024 | 0.075 | 0.221 | 0.121 | 0.136 | 0.295 | 0.251 | 0.242 | 110.00 | 8.0B | URL |
| #09 | GPT2-Medium | 2019 | 0.063 | 0.198 | 0.104 | 0.067 | **0.358** | 0.186 | 0.165 | 57.33 | 354M | URL |
| #10 | Orca-2 | 2023 | 0.103 | 0.270 | **0.161** | 0.199 | 0.330 | 0.251 | 0.271 | 177.33 | 6.7B | URL |
| #11 | Orca-2 | 2023 | 0.100 | 0.266 | 0.159 | 0.187 | 0.317 | 0.242 | 0.257 | 108.66 | 13.0B | URL |
| #12 | Deepseek-LLM | 2024 | 0.096 | 0.268 | 0.137 | 0.150 | 0.336 | 0.256 | 0.253 | 201.30 | 6.9B | URL |
| #13 | Yi-1.5 | 2024 | 0.091 | 0.263 | 0.131 | 0.136 | 0.322 | 0.229 | 0.226 | 43.66 | 6.1B | URL |
| #14 | Yi-1.5 | 2024 | 0.096 | 0.269 | 0.138 | 0.155 | 0.336 | 0.241 | 0.243 | 48.50 | 8.8B | URL |
| #15 | InternVL-2† | 2023 | 0.058 | 0.188 | 0.112 | 0.085 | 0.196 | 0.127 | 0.132 | 108.50 | 8.0B | URL |
| #16 | MiniCPM-V2.5† | 2024 | 0.046 | 0.177 | 0.085 | 0.076 | 0.254 | 0.152 | 0.122 | 51.50 | 8.4B | URL |

Table 3: Comparison of our model's performance on the IU X-ray and MIMIC-CXR datasets. The symbol † indicates that we follow the R2Gen annotation using *Findings* and evaluate with our method, as their report modifies the ground truth to an *Impression* concatenated with *Findings*. The best result is highlighted in bold, and the second-best result is underlined.

| Dataset | Methods | Publication | BLEU-1 | BLEU-2 | BLEU-3 | BLEU-4 | ROUGE-L | METEOR | CIDEr |
|---------|---------|-------------|--------|--------|--------|--------|---------|--------|-------|
| **IU X-Ray** | R2Gen | EMNLP 2020 | 0.470 | 0.304 | 0.219 | 0.165 | 0.371 | 0.187 | - |
| | R2GenCMN | ACL-IJCNLP 2021 | 0.475 | 0.309 | 0.222 | 0.170 | 0.375 | 0.191 | - |
| | PPKED | CVPR 2021 | 0.483 | 0.315 | 0.224 | 0.168 | 0.376 | 0.187 | 0.351 |
| | AlignTrans | MICCAI 2021 | 0.484 | 0.313 | 0.225 | 0.173 | 0.379 | 0.204 | - |
| | CMCL | ACL 2021 | 0.473 | 0.305 | 0.217 | 0.162 | 0.378 | 0.186 | - |
| | Clinical-BERT | AAAI 2022 | 0.495 | **0.330** | 0.231 | 0.170 | 0.376 | 0.209 | 0.432 |
| | METransformer | CVPR 2023 | 0.483 | 0.322 | 0.228 | 0.172 | 0.380 | 0.192 | 0.435 |
| | DCL | CVPR 2023 | - | - | - | 0.163 | 0.383 | 0.193 | **0.586** |
| | R2GenGPT† | Meta Radiology 2023 | 0.465 | 0.299 | 0.214 | 0.161 | 0.376 | 0.219 | 0.542 |
| | PromptMRG | AAAI 2024 | 0.401 | - | - | 0.098 | 0.160 | **0.281** | - |
| | BootstrappingLLM | AAAI 2024 | **0.499** | 0.323 | 0.238 | 0.184 | **0.390** | 0.208 | - |
| | MambaXray-VL-Base | Ours | 0.479 | 0.322 | 0.236 | 0.179 | 0.388 | 0.215 | 0.508 |
| | MambaXray-VL-Large | Ours | 0.491 | **0.330** | **0.241** | **0.185** | 0.371 | 0.216 | 0.524 |
| **MIMIC-CXR** | R2Gen | EMNLP 2020 | 0.353 | 0.218 | 0.145 | 0.103 | 0.277 | 0.142 | - |
| | R2GenCMN | ACL-IJCNLP 2021 | 0.353 | 0.218 | 0.148 | 0.106 | 0.278 | 0.142 | - |
| | PPKED | CVPR 2021 | 0.360 | 0.224 | 0.149 | 0.106 | 0.284 | 0.149 | 0.237 |
| | AlignTrans | MICCAI 2021 | 0.378 | 0.235 | 0.156 | 0.112 | 0.283 | 0.158 | - |
| | CMCL | ACL 2021 | 0.344 | 0.217 | 0.140 | 0.097 | 0.281 | 0.133 | - |
| | Clinical-BERT | AAAI 2022 | 0.383 | 0.230 | 0.151 | 0.106 | 0.275 | 0.144 | 0.151 |
| | METransformer | CVPR 2023 | 0.386 | 0.250 | 0.169 | 0.124 | **0.291** | 0.152 | **0.362** |
| | DCL | CVPR 2023 | - | - | - | 0.109 | 0.284 | 0.150 | 0.281 |
| | R2GenGPT† | Meta Radiology 2023 | 0.408 | 0.256 | 0.174 | 0.125 | 0.285 | 0.167 | 0.244 |
| | PromptMRG | AAAI 2024 | 0.398 | - | - | 0.112 | 0.268 | 0.157 | - |
| | BootstrappingLLM | AAAI 2024 | 0.402 | 0.262 | 0.180 | 0.128 | **0.291** | 0.175 | - |
| | MambaXray-VL-Base | Ours | 0.420 | 0.264 | 0.180 | 0.129 | 0.283 | 0.162 | 0.206 |
| | MambaXray-VL-Large | Ours | **0.422** | **0.268** | **0.184** | **0.133** | 0.289 | 0.167 | 0.241 |

For the X-ray medical report generation, we evaluate the model using widely used natural language generation (NLG) metrics, including **CIDEr** Vedantam et al. (2015), **BLEU** Papineni et al. (2002), **ROUGE-L** Lin (2004), and **METEOR** Banerjee & Lavie (2005). To measure the accuracy of descriptions for clinical abnormalities, we also report **Clinical Efficacy (CE) metrics**. CE metrics require the use of the CheXpert Irvin et al. (2019) toolkit to first extract labels from predictive reports and ground truth, and then to compare the presence status of important clinical observations to capture the diagnostic accuracy of the generated reports. We use **Precision**, **Recall**, and **F1** to evaluate model performance for clinical efficacy metrics. A brief introduction to these datasets and metrics can be found in our supplementary materials.

## 4.2 Comparison with SOTA Algorithms

• **Results on IU X-ray Dataset.** As shown in Table 3, it can be seen that both our MambaXray-VL-Base and MambaXray-VL-Large exhibit excellent performance on the IU X-ray dataset. Among them, the MambaXray-VL-Large model is at the SOTA level on BLEU-2 (**B2**), BLEU-3 (**B3**), and BLEU-4 (**B4**) metrics with scores of 0.330, 0.241, and 0.185, respectively. This result indicates the

Table 4: Component analysis of the key modules in our framework on MIMIC-CXR and CheXpert Plus dataset. The symbol † indicates that we are using the *Base* version of the model, while the others are the *Large* versions. **Vim-IN1K** indicates the use of weights pre-trained on ImageNet-1K; **Vim-PTD** indicates the use of weights pre-trained on 1.27 million X-ray images; **MAE** represents the Masked Auto-encoders pre-training framework; **ARG** represents the Auto-regressive Generation pre-training framework; **CTL** represents the contrastive learning loss between images and text; **SFT** represents supervised fine-tuning. **B4**, **R**, **M**, and **C** represents BLEU-4, ROUGE-L, METEOR, and CIDEr, respectively.

| Index | Vim-IN1K | Vim-PTD | MAE | ARG | CTL | SFT | MIMIC-CXR | | | | CheXpert Plus | | | |
|---|---|---|---|---|---|---|---|---|---|---|---|---|---|---|
| | | | | | | | B4 | R | M | C | B4 | R | M | C |
| #01 | ✗ | ✗ | ✗ | ✗ | ✗ | ✓ | 0.125 | 0.285 | **0.167** | **0.244** | 0.101 | 0.266 | 0.145 | 0.123 |
| #02 | ✓ | ✓ | ✓ | ✗ | ✗ | ✓ | 0.104 | 0.260 | 0.141 | 0.154 | 0.094 | 0.257 | 0.140 | 0.104 |
| #03 | ✓ | ✓ | ✗ | ✓ | ✗ | ✓ | 0.130 | 0.286 | 0.162 | 0.224 | 0.089 | 0.247 | 0.134 | 0.089 |
| #04 † | ✓ | ✗ | ✗ | ✓ | ✗ | ✓ | 0.108 | 0.264 | 0.144 | 0.170 | 0.090 | 0.249 | 0.132 | 0.103 |
| #05 † | ✓ | ✓ | ✗ | ✓ | ✗ | ✓ | 0.121 | 0.280 | 0.161 | 0.224 | 0.093 | 0.254 | 0.138 | 0.102 |
| #06 † | ✓ | ✓ | ✗ | ✓ | ✓ | ✓ | 0.129 | 0.283 | 0.162 | 0.206 | 0.105 | 0.267 | 0.149 | 0.117 |
| #07 | ✓ | ✗ | ✗ | ✓ | ✗ | ✓ | 0.105 | 0.258 | 0.139 | 0.143 | 0.082 | 0.236 | 0.126 | 0.080 |
| #08 | ✓ | ✓ | ✗ | ✓ | ✗ | ✓ | 0.130 | 0.286 | 0.162 | 0.224 | 0.089 | 0.247 | 0.134 | 0.089 |
| #09 | ✓ | ✓ | ✗ | ✓ | ✓ | ✓ | **0.133** | **0.289** | **0.167** | 0.241 | **0.112** | **0.276** | **0.157** | **0.139** |

Table 5: Comparison of text encoders used in second stage on the MIMIC-CXR and CheXpert Plus.

| LLM | MIMIC-CXR | | | | CheXpert Plus | | | |
|---|---|---|---|---|---|---|---|---|
| | BLEU-4 | ROUGE-L | METEOR | CIDEr | BLEU-4 | ROUGE-L | METEOR | CIDEr |
| Baseline | 0.125 | 0.285 | **0.167** | **0.244** | 0.101 | 0.266 | 0.145 | 0.123 |
| Llama2 Touvron et al. (2023) | 0.122 | 0.276 | 0.157 | 0.211 | 0.066 | 0.233 | 0.124 | 0.043 |
| Bio_ClinicalBERT Alsentzer et al. (2019) | **0.133** | **0.289** | **0.167** | 0.241 | **0.112** | **0.276** | **0.157** | **0.139** |

superiority of our method over other report generation methods. However, on some other metrics such as BLEU-1 (**B1**), ROUGE-L (**R**), METEOR (**M**), and CIDEr (**C**), our method does not achieve optimal performance. This reflects the need to improve the generalization of our method on other datasets.

• **Results on MIMIC-CXR Dataset.** As shown in Table 3, our method also demonstrates outstanding performance on the MIMIC-CXR dataset, surpasses all other advanced report generation methods, and achieves the most advanced level in several common indicators (e.g., BLEU-1, BLEU-2, BLEU-3, and BLEU-4). Specifically, our method improves the BLEU-4 metric by 6% compared to R2GenGPT. Encouragingly, we achieved favorable results for two of the three remaining metrics, ROUGE-L and METEOR, with scores of 0.289 for ROUGE-L and 0.167 for METEOR, which again demonstrates the superior performance of our model. In the CIDEr metric, our model achieved a score of 0.241, indicating that MambaXray-VL still has room for improvement.

• **Results on CheXpert Plus Dataset.** As shown in Table 1, our model MambaXray-VL-Large achieves state-of-the-art performance in all evaluation metric species. These include NLG evaluation metrics and CE evaluation metrics. In detail, for the NLG metrics, our scores on BLEU-4, ROUGE-L, METEOR, and CIDEr are 0.112, 0.276, 0.157, and 0.139, respectively. For the CE metrics, our scores on Precision (**P**), Recall (**R**), and F1-score (**F1**) are 0.377, 0.319, and 0.335, respectively. These experimental results fully demonstrate the superior performance of our model on the CheXpert Plus dataset. In terms of efficiency, our method took 55.18 minutes to complete the testing subset of the CheXpert Plus dataset with a parameter size of 202.32M, showing its effectiveness and efficiency in processing X-ray images.

### 4.3 ABLATION STUDY

• **Effectiveness of Autoregressive Generation for Pre-training on X-ray Image?** As shown in Table 4, we first compare the autoregressive generation (ARG) pre-training with the Masked Auto-Encoder (MAE) pre-training. From the #02 and #03 rows, it can be seen that the results achieve 0.130/0.089 on the BLEU-4 metric of the MIMIC-CXR and CheXpert Plus datasets, respectively. Note that the ARG pre-training method outperforms the MAE on all metrics, with a +45% (i.e., (0.224-0.154)/0.154) improvement on CIDEr compared to MAE. The ARG-based pre-

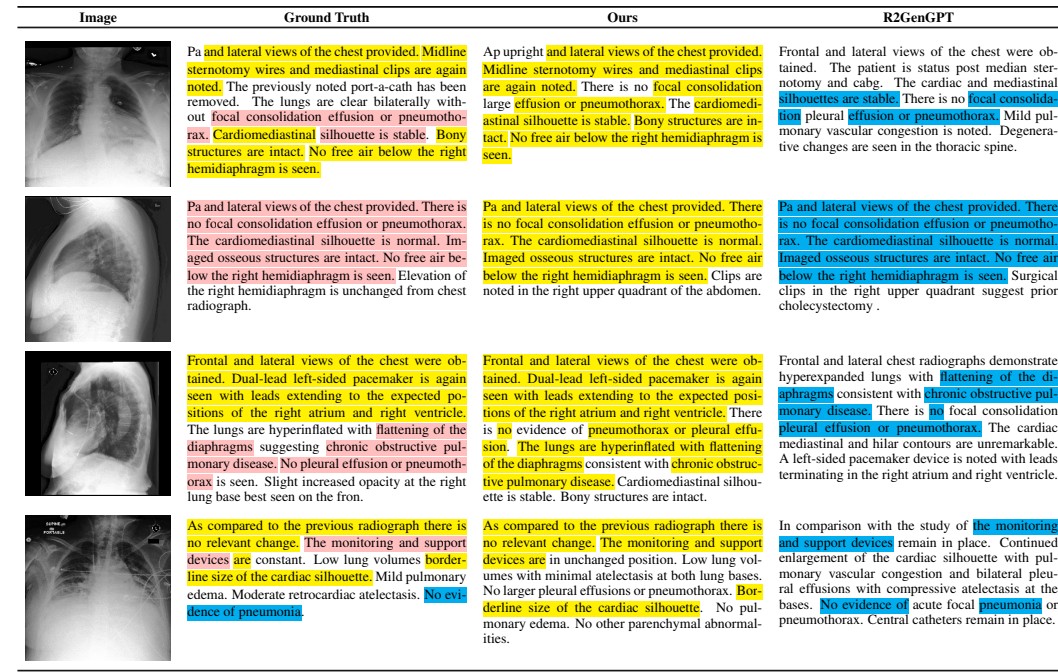

Figure 3: X-ray images and their corresponding ground-truths, along with the output of our model and R2GenGPT model generation reports on the MIMIC-CXR dataset. Matching sentences in our report are highlighted in yellow, R2GenGPT matching sentences are highlighted in cyan, and sentences matching by both models are highlighted in pink.

training achieves similar performance compared with MAE-based pre-training on the CheXpert Plus dataset.

• **Effectiveness of Xray-Report Contrastive Learning.** In addition, we further explored the impact of contrastive learning (CTL) on the final performance. The experimental results in the #05 and #06 rows of Table 4 demonstrate its effectiveness. After introducing the CTL loss, we find that the results on the MIMIC-CXR and CheXpert Plus datasets have all received improvement. More in detail, it improves the ROUGE-L metric by over +5% on the CheXpert Plus dataset. These experiments demonstrate the positive effect of the CTL loss we used in the pre-training stage.

• **Comparison between ViT and Mamba using Autoregressive Generation.** As shown in the #01 and #09 rows of Table 4, the #01 row uses a visual coder based on the Transformer architecture, while the last row uses a visual coder with auto-regressive pre-training of the Mamba architecture. It can be clearly observed that the encoder based on the Mamba architecture achieves better performance in the vast majority of metrics, both on the MIMIC-CXR and CheXpert Plus datasets, especially on BLEU-4 for the MIMIC-CXR data, where the Mamba architecture improves by +6% compared to the Transformer architecture. However, on the MIMIC-CXR dataset, the metric CIDEr does not score significantly better than the Transformer architecture. Overall, this series of experiments clearly validate the effectiveness of the auto-regressive pre-trained visual coder based on the Mamba architecture.

• **Clinical-BERT vs Llama2 in Xray-Report Contrastive Learning.** In this work, we test two models for contrastive learning in the second stage, i.e., the Bio_ClinicalBERT Alsentzer et al. (2019) and Llama2 Touvron et al. (2023). As shown in Table 5, the experimental results on both MIMIC-CXR and CheXpert Plus datasets all demonstrate that the Bio_ClinicalBERT Alsentzer et al. (2019) achieves a better performance for the X-ray report generation. We think this may be caused by the fact that the Bio_ClinicalBERT Alsentzer et al. (2019) is a LLM pre-trained using medical data, while the Llama2 Touvron et al. (2023) is pre-trained using common text data and sensitive to parameter tuning. This experiment inspired us to consider pre-training large language models by using medical data in future works.

• **Analysis on Different Configurations of Mamba Vision Encoder.** Intuitively, the large version of the Mamba model has better generalization and robustness compared to the base version, as it has deeper network layers or higher feature dimensions. As shown in Table 4, we can see that the results in lines #7, #8, and #9 (Vim-large) are significantly better than lines #4, #5, and #6 (Vim-base). Meanwhile, our Vim-large achieved optimal performance in experiments after equipping all modules. Thus, it is obvious that the larger version of Vim has a more stable performance on both MIMIC-CXR and CheXpert Plus datasets.

• **Does VLMs Pre-trained using Natural Image-Text Samples Ready for the X-ray Report Generation?** In this paper, we also conduct supervised fine-tuning on the CheXpert Plus dataset using Vision-Language Models (VLMs), including InternVL-2 Chen et al. (2023b) and MiniCPM V2.5 Yao et al. (2024). We replace the vision and language backbone network of R2Gen-GPT using the VLMs to adapt them for the X-ray image based report generation task. As illustrated in Table 2, we can find that the performance of the two models is not as good as the compared models. These experiments demonstrate a large gap between pre-training on the natural and X-ray images. In our future works, we consider further adapting the pre-trained VLMs using natural images to the X-ray image domain to achieve a better performance.

## 4.4 Visualization

As shown in Fig. 3, we give some examples to illustrate the effectiveness of our proposed MambaXray-VL model for the X-ray image based report generation. For specific X-ray images, we compared ground truth with the report generated by the MambaXray-VL model and the report generated by the R2GenGPT model. The X-ray images we chose contain both front and side views, normal images, and images containing lesion areas, enabling a more comprehensive and rational visualization. For a more intuitive visualization, we have highlighted the parts that match the ground truth. The yellow highlighted area is the part of the report generated by our model that matches the ground truth, and the blue highlighted area is the part of the report generated by the R2GenGPT model that matches the ground truth. The pink highlighted area is the portion of the report generated by both our model and the R2GenGPT model that matches the ground truth. It is clear that the report generated by our model is closer to the real report than the that generated by the R2GenGPT model, which indicates that our model is effective.

## 4.5 Limitation Analysis

This paper provides a comprehensive benchmark for the X-ray image based medical report generation, which covers the mainstream MRG models and LLMs. The LLMs evaluated in this work focus on 7B and 13B which is hardware friendly, and the LLMs with more parameters are not discussed due to the limited computational resources. The other Vision-Language Models (VLMs) developed for natural images are not benchmarked, due to the limited performance of the X-ray based MRG.

## 5 Conclusion and Future Works

In this work, we propose to benchmark the CheXpert Plus dataset by re-training the mainstream X-ray report generation models and large language models. This benchmark will help identify which large models and algorithms are leading in this domain, significantly promoting academic progress and technological development. In addition, we also propose a new Mamba-based vision-language large model for the X-ray image based medical report generation. It involves three pre-training stages which make full use of auto-regressive generation loss, Xray-report contrastive learning, and supervised fine-tuning. We validate the effectiveness of our proposed pre-trained large model on IU X-ray, MIMIC-CXR, and CheXpert Plus datasets. From the newly built benchmark, we can find that the current large language models still perform poorly on the report generation task. In our future works, we will consider introducing structured knowledge graphs into the large language model to guide the report generation.

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
