# CXPMRG-Bench: Pre-training and Benchmarking for X-ray Medical Report Generation on CheXpert Plus Dataset

## 1 Related Work

In this section, we will review the related works on X-ray Medical Report Generation, Pre-trained Large Models, and State Space Models. More works can be found in the following surveys Wang et al. (2024e; 2023a); Hartsock & Rasool (2024).

### 1.1 X-ray Medical Report Generation

In recent years, X-ray medical report generation has garnered increasing attention. To enhance model performance, researchers have pursued various improvements in different directions. Specifically, DCL Li et al. (2023) introduces a Dynamic Graph at the visual features of medical images, leveraging knowledge to strengthen the feature representation of these images. RGRG Tanida et al. (2023) takes a novel approach by using object detection methods to extract lesion regions and then generating text based on these extracted regions, ultimately combining all the text to form the final report. HERGen Wang et al. (2024a) discovers the historical information between medical reports, treating all reports of a patient as a temporally ordered whole. This approach effectively integrates the temporal and causal information of the reports. R2GenGPT Wang et al. (2023b) replaces the decoder part of the traditional medical report generation framework with a more powerful large language model, achieving improved performance. R2GenCSR Wang et al. (2024c) is a recently proposed LLM-based framework for X-ray MRG which employs the Mamba as the visual backbone and retrieves contextual samples from the training set to enhance feature representation and discriminative learning.

It is evident that the vision encoders used in these models are all conventional networks pre-trained on ImageNet Russakovsky et al. (2015): DCL Li et al. (2023) employs ViT Dosovitskiy et al. (2021), RGRG Tanida et al. (2023) uses ResNet50 He et al. (2016), HERGen Wang et al. (2024a) utilizes CvT Nicolson et al. (2023), R2GenGPT Wang et al. (2023b) incorporates SwinTransformer Liu et al. (2021), and R2GenCSR Wang et al. (2024c) leverages VMamba Liu et al. (2024). These encoders, pre-trained on non-medical X-ray images, exhibit certain limitations when extracting features from medical X-ray images. In contrast, our proposed MambaXray-VL is pre-trained on millions of datasets and has a natural advantage in the extraction of features from medical images, especially in the task of medical report generation.

### 1.2 Pre-trained Large Models

The pre-trained language models, vision models, and vision-language models are widely exploited in nowadays. Currently, the widely used MAE He et al. (2022) (Masked Autoencoders) is a self-supervised learning method for computer vision, known for its scalability and simplicity. Recently, Apple's team proposed AIM El-Nouby et al. (2024), a series of vision models using autoregressive objectives for pretraining, inspired by large language models, demonstrating similar scaling properties. ARM Ren et al. (2024) is a new self-supervised visual representation learning method based on AIM El-Nouby et al. (2024) and Mamba Gu & Dao (2023). Through the autoregressive generation based pre-training, the visual capabilities of the Mamba model can be significantly enhanced, outperforming other training strategies in terms of both efficiency and performance. CLIP Radford et al. (2021) (Contrastive Language-Image Pre-Training) jointly trains image and text encoders using contrastive learning. The key idea is to enable the model to understand and process multi-modal data

(images and text) through joint training. Inspired by these works, our newly proposed MambaXray-VL utilizes autoregressive generation based pre-training, and CLIP pre-training can achieve better results on medical report generation.

### 1.3 STATE SPACE MODEL

Since its introduction in 2017, Transformer Vaswani et al. (2017) has quickly become the preferred model framework for researchers due to its strong performance. However, as the model scales and sequences become longer, its limitations have surfaced. One major drawback is the quadratic growth in computational complexity of the self-attention mechanism with increased context length. Mamba Gu & Dao (2023) addresses these issues by using Selective State Space Models (SSMs) to improve traditional state space models and incorporating a hardware-aware parallel algorithm for recurrent operations. Vim Zhu et al. (2024) (Vision Mamba) is the first SSM model adapted for vision tasks. It uses positional embeddings and bidirectional state space models to achieve high performance, particularly on high-resolution images. VMamba Liu et al. (2024) extends Mamba by providing a global receptive field with linear complexity. MambaMLP Ren et al. (2024) is a new architectural component based on Mamba, designed to enhance feature mixing and representation learning by combining Mamba with an MLP, thereby improving performance on visual tasks. The new SSD (State Space Duality) algorithm proposed by Mamba-2 Dao & Gu (2024) can fully utilize matrix multiplication units on modern hardware, making it 2-8 times faster than the vanilla Mamba. The successful applications of the Mamba in many computer vision tasks Wang et al. (2024f); Huang et al. (2024); Wang et al. (2024b) inspired us to adapt it to the pre-trained X-ray large model for medical report generation.

## 2 IMPLEMENTATION DETAILS

• **Pre-training Stage.** Both MambaXray-VL-Base and MambaXray-VL-Large were pre-trained for 100 epochs, with batch sizes set at 256 and 128, respectively. The base learning rate, based on a batch size of 256, was set to 1.5e-4. We adopted a cosine decay schedule with a warm-up for 5 epochs and used the AdamW Loshchilov & Hutter (2019) optimizer with a weight decay of 0.05. The resolution of input images is resized to $192 \times 192$ in the pre-training phase.

In the second stage, we utilized a vision-text contrastive learning pre-training method to train MambaXray-VL, enabling alignment to the text feature space. Specifically, we used a dataset of 480,000 image-text pairs, composed of publicly available datasets from MIMIC-CXR Johnson et al. (2019), CheXpert Plus Chambon et al. (2024), and IU-Xray Demner-Fushman et al. (2016). Inspired by ARM Ren et al. (2024), we used a unidirectional scanning approach in the first stage that fits the autoregressive generation to achieve more efficient pre-training. In the second stage, we extend the scanning block to four copies in order to improve the performance of the model. During this stage, we chose to pre-training for 50 epochs, with a batch size set to 192. The visual encoder was Vim Zhu et al. (2024), loaded with weights from the first stage of pre-training, while the text encoder was Bio_ClinicalBERT Alsentzer et al. (2019), both encoders were set to be trainable. We employed the same optimizer as in the first stage, but the input image size was changed to $224 \times 224$.

• **Fine-tuning Stage.** In the downstream fine-tuning stage, we tested the model's performance on three different public datasets. On the IU-Xray Demner-Fushman et al. (2016) dataset, we set the maximum training epochs to 30 and the batch size to 20. The visual encoder used was Vim Zhu et al. (2024), loaded with weights from the second stage of pretraining, while the large language model was Qwen-1.5-1.8B et al. (2023), with *max_length* set to 60 and a validation frequency of 1, meaning we validated after each training epoch. On the MIMIC-CXR Johnson et al. (2019) and CheXpert Plus Chambon et al. (2024) datasets, we set the maximum training epochs to 6 and the batch size to 18. The visual encoder remained unchanged, while the large language model used was Llama2-7B Touvron et al. (2023), with *max_length* set to 100 and a validation frequency of 0.5, meaning we validated at both the end of each training cycle and after the training was complete. We froze the large language model and trained only the visual encoder and the visual mapper layer, by following the R2GenGPT Wang et al. (2023b).

# 3 EXPERIMENTS

## 3.1 DATASET

In the first stage of autoregressive pre-training, we used about 1.27 million medical chest X-ray images proposed in the work Wang et al. (2024d). In the second stage of image-text contrastive learning pre-training, we used a combination of training data from the **MIMIC-CXR** Johnson et al. (2019), **CheXpert Plus** Chambon et al. (2024), and **IU X-ray** Demner-Fushman et al. (2016) datasets, totaling 480k image-report pairs. Note that the CheXpert Plus dataset used here consists of images and impressions, not the image and findings combination used in the third stage. We strictly excluded any testing samples used in the third stage, resulting in a total of 210k image-impression pairs. In the third stage, We evaluate the performance of our model on three datasets, including IU X-Ray Demner-Fushman et al. (2016), MIMIC-CXR Johnson et al. (2019), and CheXpert Plus Chambon et al. (2024) dataset. A brief introduction to these datasets is given below.

• **IU X-ray Dataset** Demner-Fushman et al. (2016) [1] published in 2016 is one of the most frequently used publicly available medical image datasets for medical report generation. It contains 7,470 images and 3,955 radiology reports, with each report associated with either frontal or both frontal and lateral view images. Each report is divided into four sections: Indication, Comparison, *Findings*, and *Impression*. For a fair comparison, we used the same dataset split protocol as R2GenGPT Wang et al. (2023b), dividing the dataset into training, testing, and validation sets with a ratio of 7:1:2.

• **MIMIC-CXR Dataset** Johnson et al. (2019) [2] is one of the largest publicly available chest X-ray datasets, containing free-text radiology reports. These records from 2011-2016 include 377,110 radiographic images and 227,835 radiology reports collected from 65,379 patients at the Beth Israel Deaconess Medical Center Emergency Department in Boston, Massachusetts. For fair comparison, we used the same dataset split protocol as R2GenGPT, with 270,790 samples for training the model, and 2,130 and 3,858 samples for validation and testing sets, respectively.

• **CheXpert Plus Dataset** Chambon et al. (2024) [3] is a new radiology dataset designed to enhance the scale, performance, robustness, and fairness of deep learning models in the field of radiology. This dataset includes 223,228 chest X-rays (in DICOM and PNG formats), 187,711 corresponding radiology reports (de-identified and parsed into 11 sections), de-identified demographic data from 64,725 patients, 14 chest pathology labels, and RadGraph Jain et al. (2021) annotations. For a fair comparison, we followed the dataset split protocol used in R2GenCSR Wang et al. (2024c) which adopted *Findings* as the ground truth and split the training/validation/testing subset based on the ratio 7:1:2. The training subset with 40,463 samples, the validation subset with 5,780 samples, and the testing subset with 11,562 samples.

## 3.2 EVALUATION METRIC

For the X-ray medical report generation, we evaluate the model using widely used natural language generation (NLG) metrics, including **CIDEr** Vedantam et al. (2015), **BLEU** Papineni et al. (2002), **ROUGE-L** Lin (2004), and **METEOR** Banerjee & Lavie (2005). More in detail, CIDEr Vedantam et al. (2015) evaluates text through TF-IDF weighted n-gram matching, placing greater emphasis on the importance of words; BLEU Papineni et al. (2002) evaluates text quality through n-gram matching; ROUGE-L Lin (2004) evaluates text using the longest common subsequence; METEOR Banerjee & Lavie (2005) improves upon BLEU by considering synonyms and word order.

To measure the accuracy of descriptions for clinical abnormalities, we also report **Clinical Efficacy (CE) metrics**. CE metrics require the use of the CheXPert Irvin et al. (2019) toolkit to first extract labels from predictive reports and ground truth, and then to compare the presence status of important clinical observations to capture the diagnostic accuracy of the generated reports. We use **Precision**, **Recall**, and **F1** to evaluate model performance for clinical efficacy metrics.

---

[1] https://iuhealth.org/find-medical-services/x-rays
[2] https://physionet.org/content/mimic-cxr/2.0.0/
[3] https://github.com/Stanford-AIMI/chexpert-plus