# OpenReview forum: "CXPMRG-Bench: Pre-training and Benchmarking for X-ray Medical Report Generation on CheXpert Plus Dataset"
_ICLR.cc/2025/Conference — ICLR 2025 Conference Withdrawn Submission_

### Official Review · Reviewer_cvhC · 2024-11-01

**Soundness:** 2
**Presentation:** 4
**Contribution:** 2
**Rating:** 3
**Confidence:** 5

**Summary:**

This paper examines X-ray image-based medical report generation (MRG) by benchmarking existing MRG and large language models on the CheXpert Plus dataset, with the aim of supporting future research. Besides, the authors also propose a multi-stage pre-training strategy that integrates Mamba-based self-supervised autoregressive generation, X-ray-report contrastive learning, and supervised fine-tuning. The framework is evaluated on three public datasets: CheXpert Plus, MIMIC-CXR, and IU-Xray. However, the study primarily compiles existing models without introducing major innovations.

**Strengths:**

1. This paper provides a comprehensive benchmark for the newly released CheXpert Plus dataset, covering 19 medical report generation algorithms, which is worth encouraging.
2. The authors support their claims with both qualitative and quantitative results, adding credibility to their findings.
3. The paper is well-written, with a clear and logical structure, making it accessible and easy to understand.

**Weaknesses:**

1. Limited contribution: Benchmarking existing methods on a new dataset is not sufficient to justify publication in ICLR. Additionally, the pre-training-based framework proposed for report generation is not novel, as similar approaches have already been explored in works like CXR-CLIP and PTUnifier. Moreover, the effectiveness of the introduced Mamba network is not adequately validated (see point 4 for details).
2. Unfair comparison: The comparisons in Tables 1 and 3 are potentially biased. The proposed framework uses significantly more training data than the baselines, including 1.27 million images in the first stage and three datasets for pre-training in the second stage, whereas most of the baseline methods only train on their own datasets. This raises uncertainty about whether the performance gains are due to the proposed model or the use of additional data.
3. Lack of comparison with VL pre-training-based methods: Although the authors employ a pre-training framework, they do not compare their approach with other existing vision-language pre-trained methods for report generation, such as CXR-CLIP and PTUnifier.
4. Insufficient validation of Mamba network: The effectiveness of the Mamba network is not convincingly demonstrated in Table 4. Simply comparing rows 01 and 09 does not validate Mamba's effectiveness, as row 09 differs from row 01 not only in network structure but also in the use of additional data and pre-training strategies. A fair comparison of Transformer, CNN, and Mamba under identical conditions is needed.
5. Inconsistent performance of ARG Module: In Table 4, the proposed ARG outperforms existing MAE on MIMIC-CXR but shows a performance drop on the CheXpert Plus dataset, raising questions about the consistency and effectiveness of this module.
6. Table 3 lacks a comparison of clinical efficacy metrics, which would provide valuable insights into the real-world applicability of the models.
7. In the third stage, the authors only use "image and findings" and omit "impressions," which is an essential part of medical reports. The rationale behind this decision is unclear.

**Questions:**

Please refer to Weaknesses

---

### Official Review · Reviewer_otzk · 2024-11-01

**Soundness:** 2
**Presentation:** 3
**Contribution:** 1
**Rating:** 5
**Confidence:** 4

**Summary:**

The paper focuses on advancing the field of automated medical report generation (MRG) from X-ray images, a crucial task that can reduce diagnostic workload and improve patient care. This work addresses limitations in existing MRG datasets and models by introducing a comprehensive benchmark using the CheXpert Plus dataset. This benchmark, called CXPMRG-Bench, evaluates 19 X-ray report generation algorithms, 14 large language models (LLMs), and 2 vision-language models (VLMs) to establish a comparative framework for MRG algorithms.

The paper also introduces the MambaXray-VL model, which utilizes a novel multi-stage pre-training approach with self-supervised autoregressive generation, contrastive learning, and supervised fine-tuning, leading to significant improvements in encoding and aligning X-ray image features with text data. Extensive experiments demonstrate that MambaXray-VL outperforms existing models across multiple metrics, setting a new benchmark for MRG performance on CheXpert Plus, IU X-ray, and MIMIC-CXR datasets. The paper concludes with plans to enhance model performance by integrating knowledge graphs and expanding pre-training on medical-specific data.

**Strengths:**

The work offers a novel benchmark, CXPMRG-Bench, on the newly released CheXpert Plus dataset, representing a significant addition to the resources for X-ray-based MRG. This benchmark fills an existing gap by providing a comprehensive, standardized evaluation of MRG models, which previously faced challenges due to the lack of available comparative methods.

The introduction of the MambaXray-VL model with a multi-stage pre-training approach is innovative. By combining self-supervised autoregressive generation, X-ray-report contrastive learning, and supervised fine-tuning, the paper proposes a unique training pipeline that is highly tailored to medical image processing requirements.

The significance of this paper lies in its potential to standardize evaluations in the field of MRG, a domain where objective and comparative assessments have been historically challenging due to limited datasets and benchmark standards.

By addressing the limitations in computational costs and training stages of previous models, the proposed MambaXray-VL model could streamline the development of high-performance MRG systems that are more efficient and accessible, thus lowering the barrier for implementing these solutions in clinical settings.

The work’s impact extends to future research directions, as it suggests integrating structured medical knowledge and specialized LLMs for further improvements, signaling potential advancements in both performance and interpretability for medical applications.

**Weaknesses:**

The paper could provide a more balanced perspective by discussing the limitations of the CheXpert Plus dataset and how they might affect benchmark results.

The paper does not present the separate performance results of Stage 1 (autoregressive learning) and Stage 2 (contrastive learning), nor does it compare these with other models in terms of regression/comparative learning effectiveness.

There is a lack of detailed information regarding the selection of hyperparameter configurations used in training.

Although the paper includes benchmark comparisons, it would benefit from a deeper analysis of MambaXray-VL’s computational efficiency and accuracy compared to state-of-the-art (SOTA) models.

The paper focuses on 3D modeling, but a comparative analysis with existing 2D models for similar tasks could better highlight the specific strengths and limitations of the 3D approach.

**Questions:**

Could the authors elaborate on any inherent limitations of the CheXpert Plus dataset and discuss how these might impact the benchmark results? For example, are there any biases or data distribution issues that could influence model performance?

Could the authors provide more detailed results or ablations for each pre-training stage (Stage 1 and Stage 2)? How does each stage individually contribute to the final performance of MambaXray-VL?

Could the authors provide more insights into the selection process for hyperparameter configurations? Were there specific tuning processes or criteria used to arrive at the final values?

How does MambaXray-VL compare with other state-of-the-art (SOTA) models in terms of computational efficiency and accuracy? Are there specific metrics or settings where it particularly outperforms or underperforms?

The paper emphasizes the 3D modeling approach of MambaXray-VL, but how does it compare with traditional 2D models for similar tasks? Are there specific scenarios where 3D models outperform 2D, or vice versa?

The authors mention plans to enhance the model with knowledge graphs and domain-specific LLMs. Could they share more details on how these additions might be implemented or what challenges they foresee in integrating such components?

---

### Official Review · Reviewer_6zRq · 2024-11-02

**Soundness:** 2
**Presentation:** 3
**Contribution:** 2
**Rating:** 5
**Confidence:** 3

**Summary:**

The paper conducts a comprehensive benchmark for the newly released CheXpert Plus dataset, named CXPMRG-Bench. Then, it presents a pre-trained large model, termed MambaXray-VL, which includes three stages: self-supervised auto-regressive generation for Mamba Vision encoder pre-training, contrastive learning and supervised fine-tuning. Through this method, the experiment is extended into IU X-ray and MIMIC-CXR datasets, testing the performance and capabilities of MambaXray-VL in generating X-ray reports.

**Strengths:**

The paper is written well and easy to unserstand. The structure of the paper is commendabe, and each section is logically organized. The paper finds the weaknesses of the benchmark dataset in MRG, and uses the latest dataset to conduct a comprehensive benchmark test. It presents Mamba based auto regressive pre-training for improving performance and reducing computing costs.

**Weaknesses:**

Despite the paper's clarity, several imprecise arguments and overstatements necessitate revision and clarification:
* Compared to other algorithms, the advantages of the innovative MambaXray-VL design in medical report generation will be expressed better if there is a visual representation. The model can be segmented to show the multi-stage visual training process, such as data preprocessing, feature extraction, model training, and validation. Alternatively, the performance comparison of MambaXray-VL model with other models can be shown in the form of bar charts or line charts (such as accuracy, recall rate, F1-score, etc.) to highlight the advantages and effectiveness of this model.
* CheXpert Plus dataset can solve the disadvantage of joint training in limited dataset, and also increase efficiency. Therefore, I hope the article can explain in more detail the reasons for choosing multi-stage pre-training rather than joint training, and give more credible and specific advantages in multi-stage pre-training.
* Different algorithms have different degrees of operation on lable truncation, or do not use truncation, so when comparing the performance of the algorithm in this paper with other algorithms, does the uniformity of truncation be considered? Did the experiment compare truncation with no truncation on its impact on model training?

**Questions:**

The paper benchmarks the newly released CheXpert Plus dataset for the X-ray image based medical generation, could the research describe CheXpert Plus dataset build process in detail?

---

### Official Review · Reviewer_uWMG · 2024-11-04

**Soundness:** 3
**Presentation:** 2
**Contribution:** 3
**Rating:** 6
**Confidence:** 4

**Summary:**

The paper presents a thorough benchmarking of several X-ray report generation models and LLMs on the CheXpert Plus dataset. It argues that this benchmark provides a solid comparative framework for future algorithms. Additionally, the paper introduces a modified version of the vision Mamba (Vim) for X-ray image report generation. The paper incorporates Mamba-based self-supervised autoregressive generation, X-ray report contrastive learning, and supervised fine-tuning in their method. The paper presents extensive experimental results and shows that the Mamba-based autoregressive pre-training encodes X-ray images well. Ablation studies demonstrate results under different scenarios.

**Strengths:**

1- The paper provides a comprehensive comparison and benchmark of the CheXpert Plus dataset, which was recently published.

2- The paper aggregates multiple technical components and shows how these results outperform many existing methods.

3- Ablation studies are extensive and sound.

4- The paper is simple to follow and read.

**Weaknesses:**

1- The technical novelty of the work is limited. However, the amount of work performed in the paper is significant.

2- The justification provided why an autoregressive method works better than other methods e.g., masked auto-encoder is rather limited.

3- Some of the benchmark methods presented in Table 1 are relatively old and several newer methods were not included. For instance, the below papers were public by June 2024 (when the dataset was made public).
https://arxiv.org/pdf/2311.18681
https://openaccess.thecvf.com/content/CVPR2024/papers/Bu_Instance-level_Expert_Knowledge_and_Aggregate_Discriminative_Attention_for_Radiology_Report_CVPR_2024_paper.pdf
https://openaccess.thecvf.com/content/CVPR2024/papers/Yan_AHIVE_Anatomy-aware_Hierarchical_Vision_Encoding_for_Interactive_Radiology_Report_Retrieval_CVPR_2024_paper.pdf
https://arxiv.org/abs/2406.04449

**Questions:**

1- Provide a better description of the technical contribution

2- Elaborate more (preferrable theoretically) why an auto-regressive approach is better than a masked auto-encoder in this problem.

3- I suggest benchmarking with a few recent approaches as shown in the weaknesses section (point 3).

---

### Note · Authors · 2024-11-13

**Comment:**

Dear AC and Reviewers:
    We have read the comments from the Reviewers and were encouraged by the positive comments on the benchmark and pre-trained model. However, the overall rating is still below our expectation, thus, we want to withdraw this paper for further polishment. Thanks.

Best Regards
Authors of CXPMRG-Bench

**Withdrawal Confirmation:**

I have read and agree with the venue's withdrawal policy on behalf of myself and my co-authors.